# Dispersion Management Nonlinear Multimode Interference Mode-Locked Ytterbium Fiber Laser

**DOI:** 10.3390/nano13030535

**Published:** 2023-01-28

**Authors:** Shan Wang, Zhiguo Lv, Jintao Qiu

**Affiliations:** School of Physical Science and Technology, Inner Mongolia University, Hohhot 010021, China

**Keywords:** dispersion management, mode-locked fiber laser, nonlinear multimode interference

## Abstract

Dispersion management plays an important role in improving the output performance of a mode-locked fiber laser. Therefore, dispersion management is carried out by introducing the grating pair in our experiment. Through adjusting the distance between the grating pair, mode-locked pulses corresponding to different dispersion regimes can be realized, which typically range from soliton state in the anomalous dispersion regime to the dissipative soliton format in the normal dispersion regime. Furthermore, tunable spectrum distribution can be achieved by adjusting two intra-cavity polarization controllers. The proposed dispersion management method complements mode-locking techniques based on nonlinear multimode interference (NL-MMI). The laser can operate with self-start mode locking stably and is useful for practice applications.

## 1. Introduction

Ultrafast fiber lasers have been developed rapidly in the fields of science, industry and healthcare because of their compactness and efficiency [1,2]. Due to high broadband gain, Yb-doped fiber has become an important element in photonic systems and has been an intensive research subject [3]. In the generation of ultrashort pulses, saturable absorbers (SAs) play a decisive role in terms of power performance, operation durability and mode-locking stability. Nowadays, many SAs have been proposed and successfully applied in mode-locked fiber lasers, such as semiconductor saturable absorber mirrors (SESAMs) [4,5,6] and low-dimensional nonlinear materials. For low-dimensional material-based SAs, the damage threshold is relatively low, and mode-locking performance decreases with time. To overcome these shortcomings, artificial saturable absorption mode-locking technologies (nonlinear polarization evolution (NPE) [7,8,9] and nonlinear optical loop mirror (NOLM) [10,11,12], etc.) have been proposed and extensively applied. These artificial SAs have high damage thresholds and are independent of operation wavelength. However, the only drawback is vulnerability to environmental factors. Recently, NL-MMI mechanisms in graded-index multimode fibers (GIMFs) have attracted increasing attention from researchers to realize passive mode-locking [13,14,15,16,17]. These theories show that the NL-MMI effects have nonlinear saturable absorption characteristics similar to those of mode-locked devices such as SESAMs when an ultrashort pulse laser propagates in SMF-MMF-SMF (SMS) structure. That is, in the nonlinear regime, self-phase and cross-phase modulation (SPM and XPM) effects alter the propagation constants of multiple stimulated guided wave modes excited by ultrashort pulses in multimode fibers (MMFs) [18], thus varying the space-time characteristics of NL-MMI composite light field at the input of single-mode fibers (SMFs). For example, in the linear transmission region, the length of MMFs corresponds to the minimum coupling transmittance of SMFs. In the nonlinear region, due to the influence of SPM and XPM effects, the composite light field of high peak power laser in MMFs can achieve maximum power coupling into SMFs in the time domain (which means that the composite light field in the space domain corresponds to the distribution of the fundamental transverse mode light field matched with the SMFs mode field). On the contrary, because the pulsed low peak power laser in the multimode fiber is similar to the linear transmission, the composite light field is still the multiple transverse modes light field distribution in the spatial domain, and the coupling power into the single mode fiber is minimum due to the mode field mismatch.

In all-normal-dispersion (1 μm waveband) [19,20,21,22,23,24,25,26,27,28] and the anomalous-dispersion domain (1.5 μm, 2 μm) [29,30,31,32,33,34], the lasers based on NL-MMI as SA have been studied by many scholars. In 2018, Tegin et al. used NL-MMI mechanism for the first time to realize mode-locking operation in the 1 μm region [19]. In order to improve the modulation depth of SA and the output performance of ultrashort mode-locked fiber lasers, many mode-locking structures have been proposed in recent years. For example, Dong proposed the use of an offset-spliced GIMFs (OS-GIMFs) structure as SA in 2019 [20], and S Thulasi proposed the use of a no-core fiber (NCF)-GIMF structure as SA in 2020 [21]. At the same time, Wang et al. used the SIMF-GIMF structure as SA and successfully output the mode-locked pulse in erbium-doped mode-locked laser for the first time in 2017 [29]. In the same year, Li et al. also used the SIMF-GIMF structure as SA and successfully output mode-locked pulse in thulium-doped mode-locked laser for the first time [18]. In these research works, there is rarely a discussion on the influence of dispersion on laser performance. However, dispersion management plays a crucial role in facilitating energy scaling and pulse shortening. Therefore, it is of great significance to study the effects of dispersion on the performance of the laser for designing high-performance femtosecond fiber lasers based on NL-MMI.

In this work, dispersion management is carried out by introducing the grating pair. The different mode-locked states ranging from soliton state in anomalous dispersion to dissipative soliton format in the normal dispersion regime have been realized experimentally. Furthermore, tunable spectrum distribution can be achieved by adjusting two intra-cavity PCs. The proposed dispersion management method complements mode-locking techniques based on MMI.

## 2. Experimental Setup

The experiment scheme of the Yb-doped fiber laser is presented in Figure 1. Two single-mode laser diodes (LDs) are combined through a PBC (polarization beam combiner) as pumping sources and then pumped into the ring cavity by a wavelength division multiplexer (WDM). A Yb-doped fiber (YDF) (Yb1200-4/125 Liekki) of 0.5 m in length is used as the gain medium. A polarization-independent isolator (PI-ISO) is used to realize the clockwise transmission of a laser signal in a fiber laser. A 10:90 optical coupler (OC) is used to output 10% laser pulses from the ring cavity. Intra-cavity dispersion management is implemented through 1000 lines/mm transmission grating pair, which is placed between two collimators. The core diameter of the collimator is 10 μm. The purpose of the collimator 1 and collimator 2 is, respectively, to maximize the efficiency of the coupling of the laser in the cavity into the grating pair and to maximize the efficiency of the laser reflection from the grating pair into the cavity. Two extruded mechanical optical fiber polarization controllers (PCs) are installed to control the optical polarization states. The SIMF-GIMF structure is placed between two PCs for the purpose of laser mode-locking operation.

Theoretically, achieving a mode-locking operation can only be done by controlling the length of the GIMF (L) to exact values (L = nLπ, where n is an odd number), but this is difficult to be precise in practice. In the experiment, we can increase the number of higher-order modes by adding a 0.5 m SIMF segment between SMF and GIMF. Bending SIMF can change the ratio of the mode field diameter of the LG_00_ mode in the GIMF to that of the fundamental mode in the SMF, so that the transmission of laser in SMF-SIMF-GIMF-SMF structure does not depend on the length of the GIMF. The SIMF has a 105 μm core diameter and GIMF has a 50 μm core diameter. The nonlinear transmittance properties of SA are measured by a, centered at 1030 nm, ultrashort pulse fiber laser with 10 ps pulse duration at 45 MHz repetition rates. The experimental values are fitted by the following formula: (1)$$T\left(I\right)=1-\Delta T\times \exp\left(-\frac{I}{I_{sat}}\right)-\alpha_{ns}$$
where T(I) is the transmittance, ΔT is the modulation depth, I is the input light intensity, $I_{sat}$ is the saturation fluence and $\alpha_{ns}$ is the non-saturable loss. Then, according to the fitting curve, the modulation depth (MD)of the SA is 5.5% in Figure 2.

## 3. Results and Discussion

We study the ytterbium-doped passively mode-locked fiber laser with a distance between the grating pair of 50 mm, which corresponds to the net dispersion in the cavity of −0.18472 ps^2^. The fundamental mode-locking operation can be obtained with the pump power of 914 mW and two appropriate PC adjustments. The characteristics of the traditional soliton are shown in Figure 3. The pulse train illustrated in Figure 3a indicates the pulse period is 29.7 ns, which matches the cavity length of 6.1 m. We measure the radio frequency (RF) spectrum of the output soliton pulse with a 0.4 MHz span in Figure 3b. The first RF peak is measured at 33.58 MHz, indicating that the Yb-doped mode-locked fiber laser works with a fundamental mode-locking state. As shown in Figure 3c, the spectrum of the output pulse with the center wavelength of 1028 nm and a 3 dB bandwidth of 2.18 nm is measured. The optical spectrum shows clearly Kelly sidebands, which are caused by the interference of pulses and continuous light in the cavity and are a typical feature of solitons in anomalous dispersion fiber lasers. Figure 3d shows the autocorrelation trace of the soliton with a pulse width of 1.152 ps without external compression.

We compare the output performance of dispersion management Yb-doped mode-locked fiber laser proposed with that of other researchers using NL-MMI as SA from the aspects of center wavelength, 3 dB bandwidth, modulation depth, pulse width, repetition rate and output power, as shown in Table 1. We study the effect of dispersion on the performance of the laser for designing high-performance femtosecond fiber lasers based on NL-MMI. It is obvious that a Yb-doped mode-locked fiber laser with dispersion management has a narrower laser pulse width of 1.152 ps and a higher power output of 22 mW. Therefore, the dispersion management method is highly comparable with previous reports.

In addition to the experimental study discussed above, we also studied the influence of intra-cavity net dispersion on NL-MMI mode-locked fiber laser. With the distance between the grating pair decreasing, the intra-cavity net dispersion can reach 0.0439 ps^2^ from −0.15252 ps^2^ when the orientations of two PCs are maintained. In Figure 4, the different mode-locked states ranging from soliton state to dissipative soliton format have been realized experimentally. Moreover, the pulse width gradually increases from 1.0 to 8.1 ps as the distance between the grating pair decreases.

Furthermore, the 3 dB bandwidth of tunable spectrum can be achieved from 15.7 to 17 nm with a distance between the grating pair of 14.5 mm and two appropriate PC adjustments. In Figure 5, the pulse characteristics of dissipative soliton with 3 dB bandwidth of 15.7 nm are recorded at a pump power of 522 mW. In Figure 5a, the center wavelength of dissipative soliton at 1036.5 nm. As shown in Figure 5b, the pulse width measured is 8.17 ps by a sech^2^ fit. Figure 5c illustrates the output RF spectrum of the laser. The first RF peak is measured to be 33.86 MHz. This is basically consistent with the data of 33.58 MHz appearing in Figure 3b, which shows that the laser is in a fundamental mode-locking state. Figure 5d exhibits wider RF spectra with a 500 MHz span, which further proves the high stability of laser mode-locking state.

## 4. Summary

In conclusion, dispersion management was carried out by introducing the grating pair in our experiment and we successfully studied the effect of dispersion on the performance of the laser. By slowly adjusting the distance between the grating pair, the different mode-locked states ranging from soliton state in the anomalous dispersion regime to the dissipative soliton format in the normal dispersion regime have been realized experimentally. Furthermore, we also realized that the 3 dB bandwidth of tunable spectrum from 15.7 to 17 nm by slightly adjusting two intra-cavity PCs. The ytterbium-doped fiber laser with dispersion management can operate with self-start mode locking stably and is useful for practice applications.

## Figures and Tables

**Figure 1 nanomaterials-13-00535-f001:**
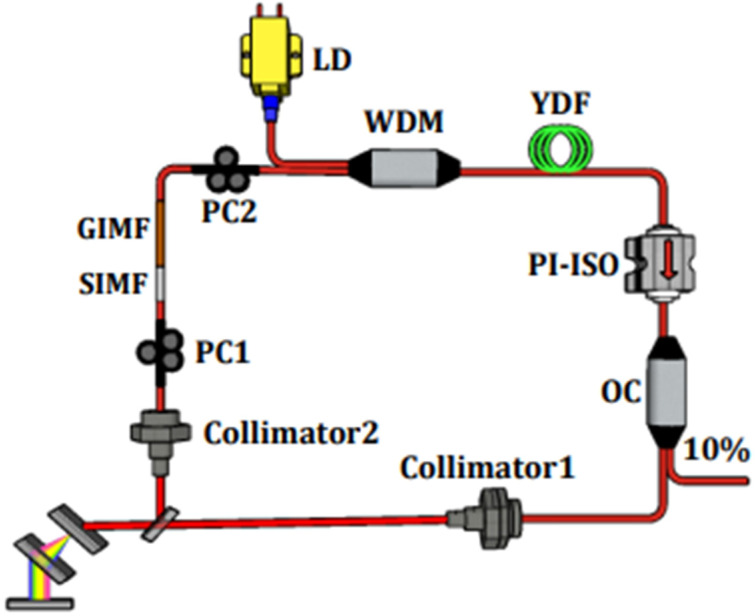
Schematic diagram of the Yb-doped fiber laser.

**Figure 2 nanomaterials-13-00535-f002:**
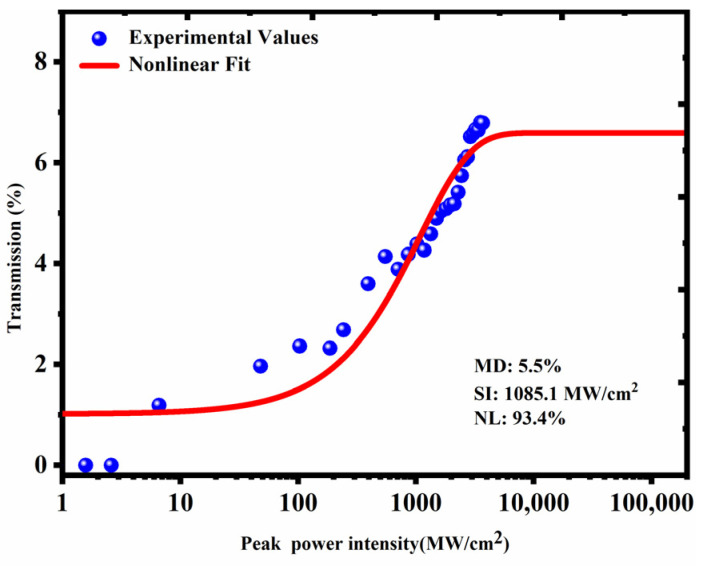
Nonlinear transmission curve of the SIMF-GIMF structure as SA. SI: saturation fluence; NL: non-saturable loss.

**Figure 3 nanomaterials-13-00535-f003:**
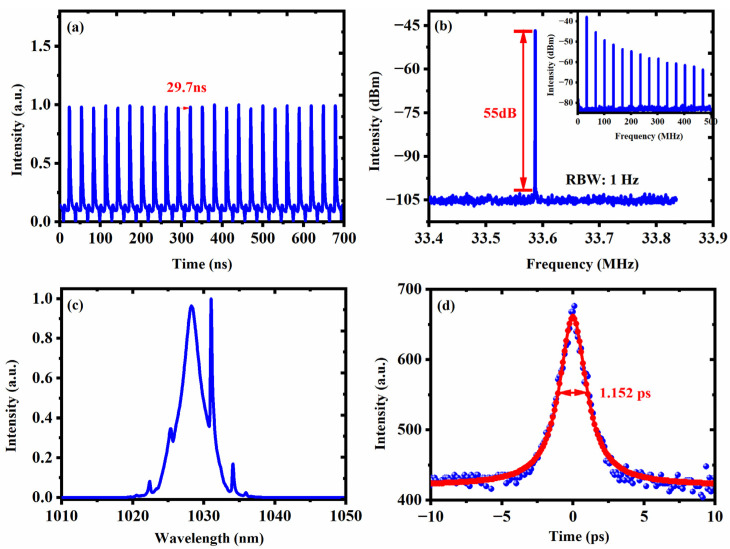
Mode-locked pulse measurements. (**a**) Optical pulse train. (**b**) RF spectrum. (**c**) Optical spectrum. (**d**) Autocorrelation trace.

**Figure 4 nanomaterials-13-00535-f004:**
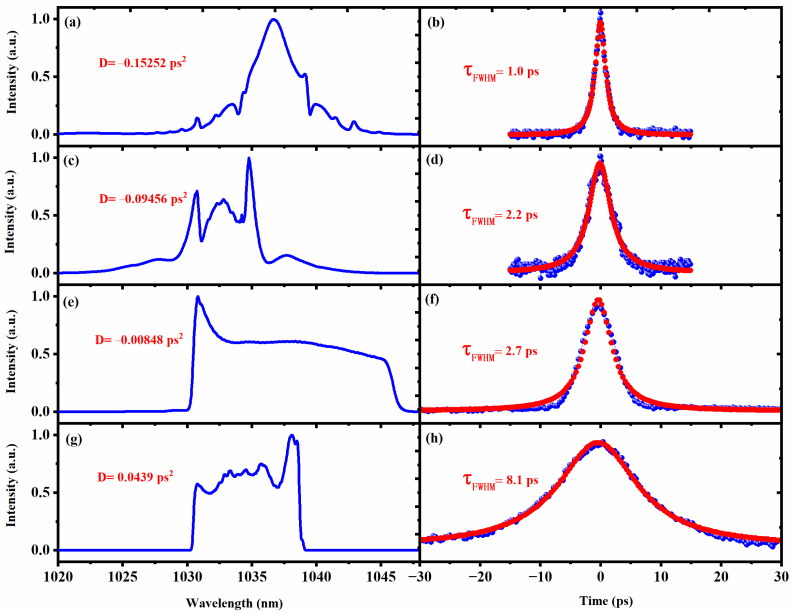
(**a**,**c**,**e**,**g**) Optical spectra at the different distances between the grating pair. (**b**,**d**,**f**,**h**) The corresponding autocorrelation traces.

**Figure 5 nanomaterials-13-00535-f005:**
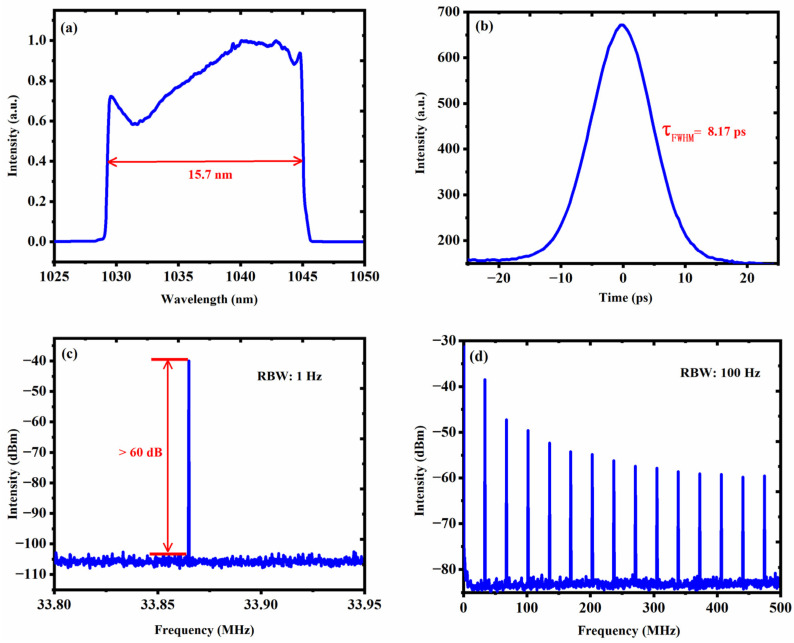
Output characteristics of dissipative solitons with a pulse width of 15.7 nm. (**a**) Optical spectrum. (**b**) Autocorrelation trace. (**c**) RF spectrum. (**d**) the RF spectrum with a span range of 500 MHz.

**Table 1 nanomaterials-13-00535-t001:** Comparison of Yb-doped mode-locked fiber lasers based on NL–MMI.

Method	Central Wavelength/nm	3 dB Bandwidth/nm	Modulation Depth/%	Pulse Width/ps	Repetition Rate/MHz	Output Power/mW	Ref.
SIMF-GIMF	1030	7.3	85	5	44.25	5.8	[19]
GIMF1-GIMF2	1040	12.3	15.28	11	27.32	3.11	[20]
NCF-GIMF	1034.2	4	24.7	2.4	21.35	2.7	[21]
GIMF1-GIMF2	1063.42	0.62	10	350	1.83	-	[22]
SIMF-GIMF	1029	7.5	-	23.9	40	37	[23]
SIMF	1045	0.6	-	180	9	1.41	[24]
GIMF	1043	9.2	50.5	26.38	27.57	7.02	[25]
GIMF	1032.3, 1054.1	5, 4	-	-	18.22, 18.23	-	[26]
GIMF	1032	5	8.8	10.67	16.67	-	[27]
GIMF-YDF	1064.14	0.04	12	105	11	-	[28]
SIMF-GIMF	1028	2.18	5.5	1.152	33.58	22	This work

## Data Availability

The data presented in this study are available on request from the corresponding author.

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
