# Peer review of "Dispersion Management Nonlinear Multimode Interference Mode-Locked Ytterbium Fiber Laser"

_nanomaterials, 2023, doi:10.3390/nano13030535_

Round 1

Reviewer 1 Report

1. The author should support his results with theoretical or analytical calculations.

2. Describe the experimental setup and how each component contributes to the setup in more detail.

3. The author should describe how much polarization he changed each time he used the polarization controller. 

4. In his experiment, the author should explain why he used step-index fiber.

Author Response

Thank you very much for your approval of our article.

Reviewer 3 Report

The authors present an interesting study on the influence of dispersion management on the output characteristics of a multimode interference mode-locked ytterbium fiber lasers. The manuscript is timely and the results can be interesting to a specialist audience. Nevertheless, the manuscript presents a few shortcomings that, in my opinion, should be addressed before it is ready for publication:

1 - The introduction, although correct, is a bit limited and should be expanded, considering the scope of the journal. See, for example, the recently published review by Y. Qi et al., "Recent research progress of nonlinear multimode interference mode-locking technology based on multimode fibers", in Infrared Physics & Technology, Volume 121, 2022, 104017.

2 - The results and discussion section should be expanded to present some general conclusions of interest, beyond the specific capacities of the presented system. What kind of dispersion management advice can be taken from these results that would be of interest for potential designers? What are the physical mechanisms at play?

3 - Although definitely readable, the manuscript would benefit greatly from some extensive English revisions.

Round 2

Reviewer 3 Report

I consider the modifications made by the authors sufficient for acceptance.